# Horses in the Early Medieval (10th–13th c.) Religious Rituals of Slavs in Polish Areas—An Archaeozoological, Archaeological and Historical Overview

**DOI:** 10.3390/ani12172282

**Published:** 2022-09-03

**Authors:** Daniel Makowiecki, Wojciech Chudziak, Paweł Szczepanik, Maciej Janeczek, Edyta Pasicka

**Affiliations:** 1Institute of Archeology, Nicolaus Copernicus University in Toruń, Szosa Bydgoska 44/48, 87-100 Toruń, Poland; 2Department of Biostructure and Animal Physiology, Wrocław University of Environmental and Life Sciences, Kożuchowska 1, 51-631 Wrocław, Poland

**Keywords:** horse, religion, early medieval, Slavs, Poland, archaeozoology

## Abstract

**Simple Summary:**

The military and prestigious importance of the horse as a cavalry mount (hussars and lancers) in post-medieval Poland has been well recorded in the pages of chronicles and historical records. However, information about the horse from Poland’s earliest Slavic history (10th–13th century) is very modest and can be found in few written sources. Therefore, to change this state of knowledge, data from archaeozoological and archaeological research were used to complement the information from chronicles. This was undertaken as part of a project focusing on the early medieval period. This article presents issues related to the role of the horse in religious rituals. This is discussed through a presentation of older and current views resulting from new research. The article presents and discusses textual and archaeological data supported by archaeozoological sources. The horse is known to be a military, elite and magical animal but also as the source of meat for food and bones for the manufacture of items. Although there is no doubt of treatment of the horse as a magical animal among the Slavs in Poland, it is clear that this animal was not buried in cemeteries in separate graves or with riders, as seen amongst contemporary neighbouring tribes.

**Abstract:**

Knowledge about horses from early medieval (10th–13th c.) Poland has been largely based on historical and archaeological data. Archaeozoological information has only been used to a limited extent. Therefore, this article aims to present the current state of knowledge on this subject, drawing on archaeozoological data from studies of horse bones. Apart from confirming earlier reflections regarding the sacred significance of the horse, additional information was obtained about specific individuals who were the subject of magical treatments. It turned out that sites with horse skeletons and skulls are few compared to the familiar presence of horse remains among kitchen waste. This contrasts with the neighbouring regions, where horses were buried more frequently among the Germans, Scandinavians and Prussians. Some new data have been obtained thanks to taphonomic analyses, which demonstrated that horse skulls of apotropaic status were not only exposed to public viewing but were also deposited under stronghold ramparts. Horses suffering from infectious diseases could also be buried under such ramparts. Considerations in the article lead to conclusions that horses were used in religious rituals as sacrificial animals, apotropaic deposits, as fortune-telling animals and cosmological figures.

## 1. Introduction 

The military and economic significance of the horse in the early Middle Ages amongst the Western Slavs has often been discussed in both the archaeological [1,2] and archaeozoological literature [3,4,5,6]. Scholars focused on the magical meaning of the horse in sacred groves and the rituals taking place there, as well as the magical meaning of horse skulls discovered in Slavic settlements. Regarding the Slavs living in present day Poland, these issues were first reported by Rajewski [7], one of the most eminent archaeologists of the older generation. He demonstrated the role of the horse in religion and magic procedures by analysing the discoveries known at the time, which consisted of figurines of horses, images of horses on vessels, weapons and combs. He also used the chronicle records from the 10th–12th century about the participation of this species in the magical rituals of the Slavs living in Western Pomerania and the Połabszczyzna (Figure 1).

Furthermore, he cited ethnographic data showing the magical position of the horse in the folk traditions of the Slavs in post-medieval times. For the first time, he analysed horse skulls, discovered during excavations being carried out at the time in early medieval strongholds. He interpreted them as a manifestation of magical and religious rituals with the participation of horses, treating them as sacrifices made before the erection of buildings and the ramparts of strongholds. Rajewski’s findings have inspired subsequent archaeologists and historians to explore this topic. It must be admitted, however, that they used the same limited number of short chronicles, ethnographic descriptions and newer, but still rarely discovered, deposits of horse skulls [8,9,10,11,12,13]. The topic of horses was discussed in analytical and synthetic monographs. Their findings, largely convergent with each other and indicating the importance of the horse in the religion of the Slavs, have permanently become part of the canon of knowledge in Polish archaeology and the history of the beginnings of Poland. After a hiatus of many years, the issue of the importance of the horse in the beliefs of the Slavs was again studied by Łukaszyk [14], using a comparative approach, drawing on the data from Scandinavia. In her book, she took into account various categories of information, including more recent discoveries of skulls and skeletons excavated from the bottom of Lake Zarańskie in the village of Żółte, surrounding a small island that was a place of special religious importance for local Slavs [15,16,17]. These new discoveries were also used in the subsequent works of the young generation of researchers interested in the religion and rituals of the Slavs [18,19,20]. In terms of sources, these publications were based on the same sources available in the aforementioned publications.

In all cases, the authors unanimously emphasized the magical meaning of the horse. The common feature of these works was that they were written by individual researchers, most often archaeologists, drawing on so-called archaeozoological expertise. For understandable reasons, they included information that could be defined as basic and understandable to them, i.e., the anatomical composition of the deposits, sex and age (life expectancy) of the animals. An unquestionable achievement of the authors was the effort to collect scattered information, most often concerning single finds of horse skulls and skeletons. For objective reasons, their publications lacked more detailed archaeozoological data obtained through taphonomic and palaeopathological analyses but also using genetic and isotopic studies or radiocarbon dating. Therefore, in this project devoted to the importance of the horse in the early Middle Ages in the Slavs inhabiting the Vistula and Oder basin, apart from collecting scattered information about this species, it was considered necessary to use the most complete set of archaeozoological methods. Among the tasks and research problems undertaken in this project, particular attention was paid to the issue of the presence of the horse in the Slavic sacrum. Already at the present stage of the ongoing research, the source information on this topic is quite substantial; therefore, the aim of this paper is to present new data, and based on it, reconsider the phenomenon of the role of the horse in the magic and religion of early medieval Slavs inhabiting the Polish lands in the 9th–13th centuries.

## 2. Materials and Methods

This article is based on historical (chronicle), archaeological and archaeozoological data. Some of this is derived from the secondary literature and some are the result of analyses carried out by the authors. Based on the literature, the most important conclusion on the economic and magical importance of the horse in the early medieval Slavs of the Polish lands is presented. This was expanded and verified with the help of archaeozoological data obtained from a project that examined the Early Medieval horse at the beginning of Poland. They are from bone assemblages of 318 sites with a chronological range from the 7th/8th but mostly from the 10th to the 13th century (Appendix A). Most of the horse bones were retrieved from the assemblages of post-consumption waste, consisting of the remains of stock mammals, game mammals, birds and fish. Only a small part constituted complete or almost complete skulls, registered at 49 sites in 38 localities, 162 in number (Figure 2). Information regarding skeletons comes from only 20 sites (Figure 3). The skeletons and skulls were subjected to macroscopic analysis, determining their anatomical composition and biological features. Age and sex were determined based on dentition using the criteria of Habermehl [21]. Measurements were carried out according to von den Driesch [22]. The height at the withers was calculated using the regression equations [23]. In the case of skeletons, skulls and individual bones to which direct access was obtained, their taphonomic features were described, particularly the presence of the traces of anthropogenic origin and the activity of predators [24]. The project also collected information on bone and tooth changes caused by diseases or using a bridle harness [25]. Due to the abundance of data in individual archaeozoological categories, only the cases were selected for this article which were considered relevant to the subject of the article indicated in the title. Taphonomic, osteometric and palaeopathological analyses are the subject of separate articles currently being prepared.

## 3. Results and Discussion

### 3.1. Archaeozoological Approaches to Early Medieval Horses

So far, the small contribution of archaeozoology to the discussion on the role of horses in the sacred sphere in the aforementioned chronological and cultural context was mainly due to the fact that the data provided are heavily dispersed in numerous archaeozoological studies, most often published in the *Annals of Agricultural Sciences of the University of Agriculture in Poznań* [25,26,27,28], in archaeological journals [29], in the chapters of collective monographs [30,31,32,33,34,35] or monographs in the strict sense of archaeozoology [6,36,37,38,39,40]. The horse was presented there as one of the many mammals whose remains were discovered during excavations. Hence, in order to revise the current state of reflection on this topic, in the aforementioned project, new research began with the creation of the most complete catalogue of horse remains, along with their biological features recorded in specific individuals. It was considered that these features were perceived by humans in living horses primarily in pragmatic terms, and some of the traces observed on skeletal elements arose as a result of deliberate manipulations, as well as the unconscious effects of using the animal, which can be best understood by examining bone lesions [41,42].

It is widely accepted that among farm mammals, from the beginning of domestication, the horse was considered a unique animal, requiring specialized breeding skills from humans to enable its effective use. In practice, not only for prehistoric peoples and the world of ancient civilizations, but also in the Middle Ages, it served, amongst others, for military purposes and transport, as well as emphasizing the prestige and social status of an individual [43,44,45]. All of these functions resulted in a positive social valorisation of exterior features (appearance), which probably meant that it was considered an appropriate representative of humans in mediation with the sphere of sacrum. Despite this, it was also a source of food raw materials for the Slavs, i.e., meat and milk, which is indicated in the first case by archaeozoological data, including traces of slaughtering activities [6,35], and in the second by historical records of milk consumption [46,47]. It was also used to obtain secondary raw materials, such as leather [48] and bones, for the manufacture of utility items [49,50]. 

### 3.2. The Horse as a Medium between Deities and the Slavs

Thanks to comparative studies and general humanistic reflections, as well as the analysis of written sources and ethnohistorical information on traditional communities, such as the early medieval Slavs, it is possible to assume that in social life the sacred sphere and the technical-utility sphere is combined, creating something that in the literature on the subject as “magical syncretism” [51,52,53], [19] (pp. 24–37). It is believed that all practical activities carried out in the socio-cultural spheres were given a religious and mythical meaning. In them, one can see a special role of the horse in the cultures of various Indo-European peoples [54], including Slavs [14], Balts [55], Germans [5,56] and Scandinavians [57,58]. Their patterns of perception of horses were quite universal, although researchers also see differences between individual communities.

When discussing the importance of horses in the culture of the early medieval Slavs, it is impossible to ignore written sources [46,47]. Although they are few and, in addition, include very laconic mentions, they undoubtedly represent the most spectacular for those times, refer to the most important events with the participation of horses, and on this occasion indicate those features that had a specific cultural value—of utility and magic. The oldest mentions show the multifaceted nature of the relationship between people and horses. In Procopius of Caesarea’s chronicle, there is information about the presence of members of the Slavic Sklavens and Ants, who lived on the Danube at that time, among the Roman cavalry [59] (pp. 81–89). However, these were probably elite representatives, since in another place the same author mentions that “[…] when they go to battle, most of them go on foot [...]” [60] (p. 55), [61] (pp. 222–223). The possession of horses by the Slavic elders is reported in the 7th century in the work *Theophylact Simocatta* [62] (p. 76), [60] (pp. 68–69). Belonging to the elite of horse warriors and the military importance of the cavalry was also indicated by Ibrahim ibn Jakub, around the middle of the 10th century, describing the principality of the first ruler of Poland, Mieszko I, as follows: “[...] he gives these men clothes, horses, weapons and everything they need. And when one of them has a child, he orders him to pay his salary from the moment of birth [...]” [63] (p. 50). In addition to the prince’s superior role in purchasing expensive horses and handing them over to his warriors, the record also preserves the care that the members of this elite formation do not run out of anything, including the means to raise children. This concern could have been a kind of princely commitment to his warriors for their service and sacrifice [64]. The multitude of horses was to be characteristic of the Nakona principality, which in the 10th and early 11th centuries covered part of the territory of the Polabian Slavs. According to Ibrahim ibn Jacob, “His lands [are] cheap in price, and horses abound. [They] are exported from there to other countries” [47] (p. 147). The horse trade is also confirmed by the customs tariff from Raffelstetten on the Danube from 903/6, where these animals are found next to slaves as the main subject of trade [65] (pp. 460–461), [66] (pp. 45–47). Subsequent mentions of the importance of this species and the perception of it as a unique mammal refer to the Great Moravian state (early 9th–early 10th century). According to ibn Rosteh, all the mounts belonged to Świętopełk I of Great Moravia, who was supposed to eat only horse meat and horse milk [46] (p. 453). Subsequent written sources refer to the role of horses in funeral rituals. In “Golden meadows” by historiographer al-Mas’ūdī, written around 947, a description of a funeral among the people of S[a]rbin, which can be associated with some West Slavic tribes, is quoted. According to this colourful description of the funeral of a ruler or leader, his body was burnt on the pyre together with the horses [67] (pp. 151–152).

Another source of information about horses is contained in the chronicles of the fortune-telling (hippomancy) ceremonies taking place in temples. Their authors describe the presence of large males with uniform white or black coats [10,47]. A black-coated Szczecin mount of Trzygław was mentioned by Herbord in a chronicle prepared by him in the second half of the 12th century, describing the life of Otto of Bamberg in the years 1124–1128 and his Christianization mission in Pomerania: “[...] they had a horse of extraordinary size, fat, black and very wild; throughout the year it was free from activities, and it was so holy that no one was worthy to ride on it, but it had one of the four priests of the temples as a zealous caretaker […]” [47] (p. 174). This animal had such a high status that a separate priest was appointed to care for it, who was perhaps also responsible for the extremely valuable equestrian equipment intended for this horse [68].

Another source, this time relating to the Arkona cult of Świętowit, is Saxo Grammaticus’ chronicle from the end of the 12th and the beginning of the 13th century [10]. There is the following sentence in it: “Besides, it [deity] had a separate horse of white coat, plucking hair from its mane or tail was considered an impious act. Only the priest was allowed to give it an edge and mount it, lest the horse become common through frequent use. On this horse, as it was commonly believed in Rügen, Świętowit-that was the name of the deity-fought against the enemies of his holiness” [69],47] (p. 180). The horse was also used for fortune-telling with the use of crossed spears stuck in the ground; in this case it was supposed to be three rows of two spears, which for us, representatives of modern society, appears to be an unusual and intriguing custom. It should be remembered, however, that despite the great reverence of divine mounts, the specific deities who rode them, and not the mounts themselves, were the object of religious worship [18] (pp. 110–111). On the other hand, it was the horse that was necessary to worship the deity, and thus became as important as the priest caring for it and probably more important than an average member of the Slavic tribe.

Another example of the significance of the horse in the sacred sphere is provided in the chronicle compiled by Thietmar (Bishop of Merseburg) in 1012 [70]. In the part describing the temple in Radogoszcz and the fortune-telling taking place in this town, he made the following record, according to which the priests “whispering mysterious words to each other are digging the earth with tremble in order to investigate the essence of doubtful matters on the basis of the fortunes thrown out. After the fortune-telling is over, they cover their fortunes with green turf and, having plunged two spearheads into the ground, lead the horse through it with humble gestures, whom they consider to be the greatest thing and worship it as sacred” [47] (p. 172).

### 3.3. Archaeological Contexts and Religious-Magical Meanings of Horse Bone Deposits

The above-mentioned historical information in the aforementioned project also became an inspiration to search for magical–sacral meanings of the horse in the Slavs through the archaeozoological data observed on the skeletal remains of horses. Taking into account the state of their presence seen in situ, three basic categories have been distinguished: (a) not articulated skeleton elements (whole bones or fragments); (b) whole skeletons or anatomical elements of skeletons (articulated or not articulated); and (c) skulls with mandibles, only the skulls and only the mandibles [17]. They were discovered not only in former settlements (strongholds, settlements) and cemeteries but also in early medieval ritual and cult sites [71]. Only a small part is complete or almost complete skulls, registered at 49 sites in 38 localities, 162 in number (Figure 2). Information regarding skeletons comes from only 20 sites. They were discovered in cemeteries, i.e., in Dziekanowice, Jordanów, Górzyca, Pień, and from one to a dozen or so were registered in fortified settlements, such as Gdańsk, Głuchowo, Krzyżownica, Kałdus, Opole–Ostrówek, Ostrów Lednicki and Tum (Figure 3). A special case is the incomplete skeleton of a horse from Kraków-Sukiennice, as it was discovered on the edge of a medieval necropolis or in the neighbouring settlement [72]. In a pit, an articulated skull, mandible and cervical vertebrae were discovered. Other exceptional deposits are the elements of horse skeletons, which are the result of depositing parts of horse bodies in three men’s graves in the old-Magyar cemetery in Przemyśl [73]. They include skulls, metapodia and phalanges. An example of partial horse skeletons is deposit found in Żółte. At the bottom of the Zarańsko Lake in the south of an island were registered different postcranial elements (long bones, ribs, scapula and pelvis) of at least three horses [17]. Only a mandible probably belonged to one of them. Last but not least, in spaces of strongholds and settlements, the most common are whole bones or fragmented, which may constitute food remains, as they are scattered among remains of other animals, such as cattle, ovicaprids, pig, wild mammals, birds and fish [27,29,35]. Already this short statistic seems to be a good enough indication of the potential of the source information, from which only those relating to individual deposits have been selected in order to present insights that bring new elements to the consideration of the above-mentioned issues.

### 3.4. Taphonomic Analyzes

One of the goals in taphonomic analysis is to indicate the genesis of the deposited bones [24]. The relevant findings are the result of observing the degree of degradation, mechanical damage and any modifications to the original shape (morphology) of individual anatomical elements (whole or preserved as fragments), discovered in anthropogenic or natural stratigraphic contexts.

In this project, such research included skulls and skeletons discovered in contexts considered to be ritualistic. In the case of skulls, the main aim of the research was attempting to identify traces of human activity on them, which could be the result of subsequent activities, from killing the animal to the final deposition of its head, i.e., when making a specific transformation of the animal from its utilitarian function to a deposit of ritual-magical meaning.

Previous approaches rarely dealt with these issues. An example is the indication of holes on the skulls formed after injuries, resulting in the death of one of the horses in Ostrów Tumski in Wrocław [74] and Ostrów Lednicki [6] (p. 64). This time, the project was able to examine not one but several skulls. As a result of the conducted analyses, two main categories were indicated: a) skulls with modifications and b) specimens without modifications, whether anthropogenic or environmental.

An example belonging to the first group is the skull discovered in Gdańsk at Grodzka Street, at the bottom of the timber framework of the former stronghold [75]. On the surface of the frontal and nasal bones we noticed, there were traces of glazing, and numerous oblique negatives of narrow, flat and sometimes sharp cuts/scratches of variable orientation (Figure 4a).

The specimen belonged to a male of about 20+ years of age, with traces of wear in the teeth indicating that it had worn a bit. Similar features were noted on the skull from Biskupin, which was discovered in the context of the “source of spring water” [7,76]. This skull came from a young mare aged 3–3.5 years old. There were also quite numerous scratches on the nasal bones, usually oblique, similar on both sides of the bones. The same, although longer, were on the left frontal bone, including oblong, oblique and intersecting. A completely opposite taphonomic image was recorded for the skull from the castle complex in Płock, from one of the places interpreted by Szafrański [12] (pp. 139–154) as a pagan reserve (see critical remarks [77] (pp. 63–70) and from one more skull found in the rampart of the Gdańsk stronghold (Grodzka street, site 1). Both skulls were unmodified (Figure 4c) and belonged to males about 11–12 and 11–13 old, respectively. 

On the basis of the above examples, it can be concluded that there are at least two completely different ways of treating horse heads, preceding their more or less intentional depositing. Some were most likely assembled as whole heads, while others, detached from the neck, could have undergone preparation and then have been used as pure “preparations”, i.e., without soft parts. Furthermore, taking into account the described traces, with the simultaneous lack of signs of weather effects, it can be assumed that they were initially stored in rooms with limited access to atmospheric factors, such as rainfall, sunbeams, or drastic temperature changes. Subsequently, they could serve as elements displayed for public view. An example of such activities is the aforementioned skull from Grodzka Street in Gdańsk, discovered under the construction of the road of one of the gates leading to the interior of the castle. The traces noted on the surface of the skull and the place of discovery suggest that the skull was originally placed in a room near the gate or above the gate. This would have protected the skull against excess moisture, precipitation, sunlight and temperature fluctuations, factors causing weathering and, consequently, changes on the bone surface, which had not been found. Perhaps a similar example is the specimen discovered within the Berlin stronghold of Spandau, which, according to German researchers, was originally to be placed above the gate passage [78] (p. 84). It is known from ethnographic sources that horse skulls had a protective power [79] (p. 594), [80] (p. 351). Therefore, one of the intentions of placing them within the castle structures, including over the entrance gates to the strongholds, was the belief in the protection of the closed spaces used by humans.

Finally, it is worth adding that, in Hungary, horse skulls from individuals killed by blows to the forehead were often placed in stables to prevent horse diseases or to keep the animal in a healthy condition. Horse skulls were also displayed in vineyards, stables and apiaries to protect the locations from the evil eye and omission [81] (p. 56). In this case, it would be a confirmation that there were equestrian quarters within the gates and boroughs. The custom of placing skulls on the roof slopes was immortalized in the 16th century painting by P. Bruegel the Elder, entitled St. Jerzy (Figure 5).

### 3.5. Paleopathological Analyzes

In considering the magical qualities of the horse, the significance of which was noted in the historical sources cited earlier, palaeopathological research was also useful; through this it was possible to establish certain characteristics of the animal that indicate its dual function. The first example is an individual deposited in its entirety at the bottom of one of the rampart’s timber frame boxes of the eastern part of Gdańsk’s early medieval stronghold [82]. The discovered skeleton was from a male approximately 8–9 years old. Modifications indicating the use of the animal for riding were noted on the teeth and spine. In addition, lesions caused by bacteria, such as *Brucella abortus*, *Mycobacterium bovis*, *Burkholderia malelli*, *Acranobacterium pyogenes*, *Staphylococcus* spp., as well as *Aspergillus* spp., have been noted on the skull (Figure 6) and other skeletal elements [83].

The infection can result in brucellosis, tuberculosis or glanders. After being infected with an infectious agent, the horse developed symptoms that negatively affected its efficiency and thus performance. Therefore, the question arises whether these disease symptoms, undoubtedly causing the deterioration of its condition, prompted the owners to sacrifice the animal as a “foundation offering” during the erection of the fortifications of the city in Gdańsk? It is difficult to settle this unequivocally. In response, comparative material and examples of other skeletons with traces of diseases, also buried in places considered sacred (sacrificial), may be helpful. In the case of horses, one of these is the island in Żółte on Lake Zarańskie in Western Pomerania. Around it, in the 9th–12th centuries, several horses were buried, including an individual with visible signs of disease in the joints, causing the loss of mobility [17]. It is worth considering these examples in the context of the older skeletons of this species from the periods of Roman influences and the migration of peoples, which came from individuals with visible signs of disability of the bones, which was also visible in the external appearance of living individuals. These horses were buried with men who were considered shamans [84] (pp. 290–293). With reference to the valorisation of sick/disabled animals, it is worth adding that in the population of the Przeworsk culture living in Kujawy, the bodies of dogs with visible lesions were buried within temples and homes [85]. This was revealed by palaeopathological studies carried out on nearly 111 skeletons, including 70 preserved skulls [86]. Almost all of them were diagnosed with various conditions. Therefore, it cannot be ruled out that sick animals were given magical qualities by burying them in places important for humans.

### 3.6. Horse Foundation Offerings in the Light of Archaeozoological Data

In archaeological publications, skulls and skeletons discovered in the immediate vicinity of man-made structures are quite commonly regarded as foundation offerings [7,11,12] (Figure 2 and Figure 3). This was the interpretation of the deposits from the abovementioned rampart of the stronghold in Gdańsk, the next ones from Gniezno, Czeladź Wielka, Pułtusk, Santok and Sypniewo [11]. Apart from the ramparts, the foundation sacrifices were also intended to protect residential buildings (e.g., cottages). Few such examples are known from the areas of early medieval Czersk [11], Gdańsk [9], Opole-Ostrówek [87], Wolin [7] or Kałdus [88]. Among the more recent discoveries, the protective function can be attributed to horse skulls from the settlement in Bródno Stare [89]. Magical significance was also given to the specimens lying in the well in the Sypniewo stronghold [90] (pp. 68–69) and the skulls from the Kalisz stronghold’s homestead, where one rested in the barrel, and three skulls outside [91]. Other examples of the presence of horse skulls come from underwater excavations in Żółte [15,17], Nętno [92], Chycina [93] or Rybitwy [34].

Magical meanings are given to skeletal deposits resulting from a horse’s body being buried in a separate cavity within a cemetery with human graves (Figure 3). They were discovered in the previously mentioned cemeteries in Dziekanowice [94,95], Pień [96], Kałdus [97] (p. 107), [98], Jordanów [99] and Górzyca [100]. All were in anatomical order with no evidence of carcasses processing and consumption. As this might play similar role to that in Baltic tribes, who buried horses in separate pits within commentary with human graves [101,102], it should be emphasized that a horse’s body was placed in one grave with a human [55,103,104]. If so, it can be assumed that a horse and his master (armed rider) rested in it after they both died (e.g., in battle). When the owner of the horse died, the horse was also put to death and was supposed to serve the owner in their posthumous existence. Thus, the reasons for burying horses in the cemeteries in Kałdus, Jordanów, Górzyca and Pień would be quite different from those of the Germans and Scandinavians. First, it can be assumed that these were not animals that were killed at the same time as the death of their owner—for example, an armed rider. Secondly, they were also not victims of armed battles where they died but their rider survived, because moving a horse from a battle place (especially a distant one) would be too complicated an undertaking. This means that at least some of the horses, i.e., those lying on their side, including, in particular, the Pień individual, were killed on purpose in the places where people were buried [96]. It cannot be ruled out that the bodies of dead horses were deposited in cemeteries, as it was assumed in the case of an aged (15–18 years old) individual from Dziekanowice [95]. However, in such a situation, it should be assumed that these horses, when alive, held a special value for the owner or even the local community, since their bodies were moved from the place of death (probably nearby) to the cemetery, a space of special sacred importance. Finally, it cannot be ruled out that the skeletons discovered within the cemeteries belonged to the horses buried (killed) alongside their owners, only that they were then placed in separate burial pits. In all cases, it can be considered that the location of the burials of such horses in the cemetery would be characterized by intentions of a non-utilitarian and, therefore, magical, symbolic, and at least emotional nature. On the other hand, it is worth mentioning that deposition of an animal’s body in different societies and periods could have expressed different intentions. [105,106,107,108].

A foreign category of objects involving horse bones are human skeletons discovered in the cemetery in Przemyśl. The objects found there were the basis for stating that in the graves were buried representatives of the old Magyar population [73]. This was also confirmed by recorded horse bones from the heads (skulls) and the hand and foot. Yet another case of the participation of horses in burials are the discoveries in Chodlik, where in the embankments of two mounds, apart from burnt human remains, burned horse remains were also recorded [109]. In this case, however, these are most likely indications of the burning on the pyre of a deceased person with a horse, which refers to the record made around 947 by the Arab chronicler al-Mas’ūdī, according to which the Slavic people of S[a]rbīnwould burn, together with the body of a king or a leader, also the mount of the individual [109]. The quoted author also cited examples from three cemeteries located in south-eastern Poland (Brzeźnica, Horodyszcze and Guciów), where horse remains were registered along with human remains [109].

Summarizing this part of the considerations, showing the differentiation of contexts with the deposits of skulls, skeletons and remains of horses, it can be said that they were an important component of magical rituals for the Slavs, mainly in pagan times. However, the dating of some deposits to the times after the introduction of Christianity indicates the survival of old customs despite the acceptance of the new religious worldview or can also be interpreted as a manifestation of the return of pagan practices and the weakness of Christianity in mastering the belief sphere of a large part of the Slavs, including those coming from the elite whose horses were one of the attributes, as it results from the analysis of historical data [47].

### 3.7. The Magical Meaning of the Horse Preserved in Artistic Motifs

The importance and rank of this species in the mythical reality and the symbolism of the early medieval Western Slavs can also be seen in some decorative motifs created by contemporary artists/craftsmen on elements of equestrian equipment and manufactured figurines [14,19,20,110,111,112]. An example is an iconographic representation placed on the bronze ferrule of the Oldenburg knife scabbard, dated to the 10th/11th century. It consists of a vertical axis at the ends of which there are two central figures. One (on the upper end), with an imperious pose, can be treated as the central (divine), and the lower one, with two horse heads, as a reflection of the mythical charioteer [113,114] or a three-headed deity [115]. For this attempt to include archaeozoology in reflections on the meaning of the analysed iconographic representations, figures placed symmetrically on both sides of the axis are important. They are located in the zone between two figures—one considered to be god and the other to be human [113]—or between two deities [115]. These are images of people and horses. Animals, through their position and direction, are considered to be intermediaries connecting the world of humans with deities [113]. Two of them reflect the natural vertical pose for horses, but the next one has a twisted neck so that its head is unnaturally directed back, and at the same time towards the supreme divine figure, shown in an imperious pose and located at the top of the axis (Figure 7a). The same shape was given to the necks of horses placed on zoomorphic spurs of the Lutomiersk type (Figure 7b), the best-preserved specimens of which were discovered at the necropolis in Ciepłe [19,111].

The project also includes an in-depth analysis of the arrangement of skeletons in situ, assuming that it is a derivative of the activities performed during the burial ritual, from killing to placing the body in a cavity (grave). It led to the notion that the horse buried in the Pień cemetery also had his head turned backwards by twisting his neck (Figure 7c). As the authors suppose, thanks to this position and the position of the animal on its side, access to large blood vessels running in this place was obtained [96]. This, in turn, made it possible to bleed the horse by cutting blood vessels, and consequently led to its mild, slow death. Such behaviour proves the respect that the horse chosen for sacrificial purposes was blessed with. Such behaviour is confirmed in Indo-European comparative material (religious and anthropological sources) showing the respect for the buried animal [54].

So perhaps the image on the Oldenburg ferrule should be an image of a bloody horse sacrifice, which was made in the earthly zone, to reach the heavenly realm? Even if such an interpretation may raise doubts, it cannot be denied that the inclusion in the project of the effects of observations and analyses of the in situ arrangement of elements of the horse skeleton, whose body was buried among human graves, and therefore for reasons not so much practical as religious and magical, may be a source of inspiration in explaining figural representations, the style and set of figures of which were found to be directly related to the Slavic notions of a sacred nature. Perhaps another form of offering is represented by miniature horse figurines made of copper and wood alloys (Figure 8), described in the scholarly literature [14] (pp. 122–127), [111,117,118].

The formal and contextual differentiation of the collection in question seems to be significant enough that at least some of them can be considered votive figures. Perhaps, under certain circumstances, they replaced the sacrifice of a live horse [19] (pp. 202–204), [110]. In terms of semantics, they can be seen as creations that mark the high importance of divine mounts, which, according to the chronicle records mentioned above, were kept in the largest cult centres. Schematic saddles placed on the backs of horses may be a representation of the rich saddles located in the shrine of Świętowit in Arkona, described by Saxo Grammaticus as: “[…] nearby you could see the bridle and the deity’s saddle and some other signs of divinity […]” [47] (p. 179). This deity, on a white horse, waged a battle against his enemies, and it may be assumed that his mount was equipped with the saddle in question. A saddle was also to be worn by the black mount of Trzygław, described by Herbord [119]. Showing saddles on the backs of horses could be an attempt to show the fact that the god himself rides living, life-sized animals [13] (p. 81). Perhaps in this way, the necessity of the saddle was also emphasized as a device enabling the adoption of the best ergonomic position, so important during military skirmishes, as well as decisive for the comfort of traveling on the back of an animal, which, while in motion, cyclically changes the statics of the entire torso, causing the rider/traveller to sway constantly.

## 4. Conclusions

Most of the discussed categories of horse deposits and artworks with its images come from the younger phases of the early Middle Ages, especially from the 10th–12th centuries. It is possible that this is a reflection of the complex socio-political processes taking place, where the older tribal organizational structures of the Slavs transformed into more complex states. The success of these transformations was largely determined by the introduction of cavalry into armed formations on a larger scale than it was during the tribal period, i.e., until about the middle of the 10th century. The increase in the military role of the horse and its popularization as an indispensable component of combat strength also translated into the ideological sphere, so not only did the animal become an attribute of prestige but also an indispensable means of influencing sacred powers.

The data and reflections presented above allow for the conclusion that the sacral-mythical ways of using the horse as one of the components of the domestic fauna of the time were both specific and varied. Individuals of this species could be part of religious rituals as (a) sacrificial animals, (b) apotropaic deposits, (c) fortune-telling animals and (d) cosmological figures. The above-mentioned roles should be seen as flexible, not necessarily assigned to a specific animal once and for all, although it could also be so. They could also be given to an animal depending on the specific utilitarian and/or sacred needs of a human (priest, chief, prince), and their choice depended on the animal’s biological characteristics (sex, age, body size) and behaviour (temperament). They were subject to constant evaluation and valorisation, carried out in specific and diverse cultural, social and religious contexts.

Some of the noted source collections (contexts) are still difficult to explain and require further in-depth reflection. One of them is the sparse presence of horse burials or the lack of skeletal human and horse graves in the Slavic lands of what is now Poland. In this case, only a simple conclusion is possible that the Slavs of the Vistula and Odra basins saw the importance of these animals differently in the practices accompanying the death of equestrian warriors, as well as the horses themselves, to neighbouring peoples—the Germans, Scandinavians and Prussians. What were the reasons for this? Was it too difficult to own a horse, to the point where it would become the property of another person when its master died? If the richest member of the community, with the rank of chief (prince), would buy horses and equip his warriors with them, as Ibrahim ibn Jakub states, would he not accept losing them after the death of his mounted warriors? If so, it would mean that in Poland during the Piast period (10–14th centuries), horses were, for a long time, the exclusive domain of the ruling elites (e.g., castellans).

## Figures and Tables

**Figure 1 animals-12-02282-f001:**
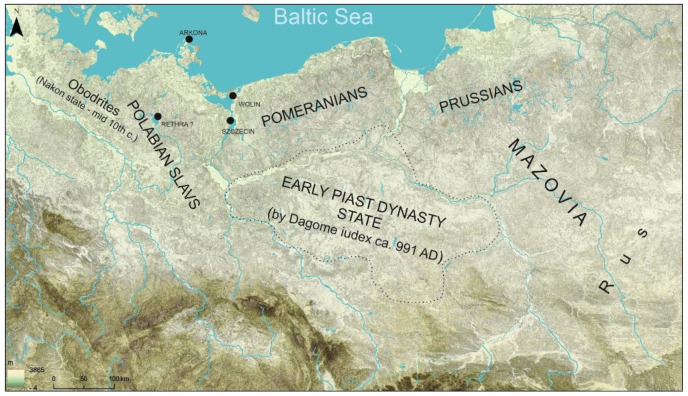
Historical regions of Western Slavs (ca. mid-10th–early 12th c.), their neighbours and the main Slavic cult temples mentioned in historical records (prepared by D. Makowiecki, drawing M. Skrzatek).

**Figure 2 animals-12-02282-f002:**
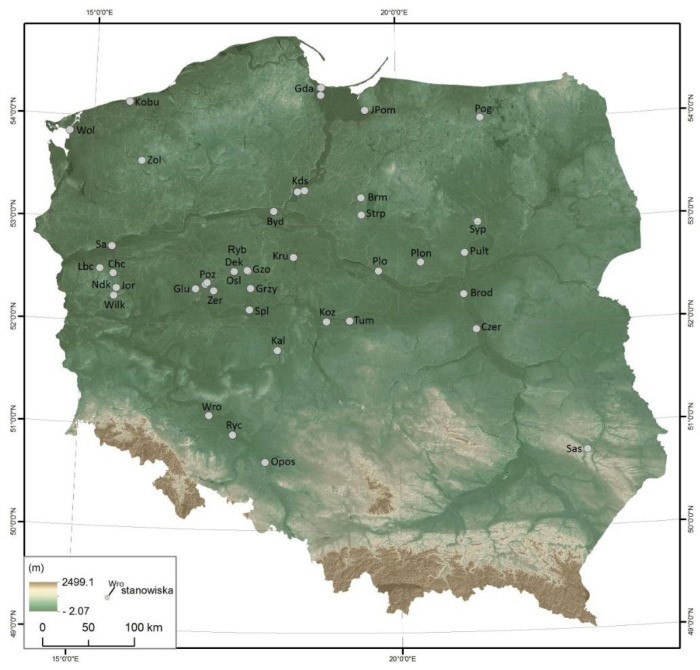
Sites with finds of horse skulls (the acronym of the site is given first, the number or name of the site in the following place, the number of skulls in brackets): Bis—Biskupin 15a—spring (2); Brm—Brodnica-Michałowo, 1 (1); Brod—Bródno Stare (3); Byd—Bydgoszcz, stan. 1; (1); Chc—Chycina, 19; (3); Czer—Czersk (1); Dek—Dziekanowice, 22 (2); Gda—Gdańsk, 1 (7), Gdańsk, 2 (2), Gdańsk, ul. Grodzka (5); Głu—Głuchowo, 1 (7); Gzo—Gniezno, 15 (1), Gniezno, 15d (3); Gniezno, 76 (2); Grzy—Grzybowo, 1 (1); Jpom—Janów Pomorski, 1 (1); Kal—Kalisz-Zawodzie (5); Kds—Kałdus, 2 (5), Kałdus, 3 (4), Kałdus, 4 (2); Kobu—Kołobrzeg-Budzistowo 1 (5); Koz—Kozanki Podleśne 1 (3); Kru—Kruszwica, 2; (1), Kruszwica, 4 (1), Lbc—Lubniewice, 9; (3); Ndk—Nowy Dworek, 27 (1); Opos—Opole-Ostrówek, 1 (4); Osl—Ostrów Lednicki–Rybitwy, 3a (6), Ostrów Lednicki, 2 (6); Plo—Płock, (1); Plon—Płońsk 7 (4); Pog—Poganowo, 4 (11); Polt—Półtusk (1); Poz—Poznań, ul. Wodna 13 (2); Pow—Poznań-Wilda 164 (2); Ryb, 3 (1); Ryc—Ryczyn, 1 (3); Sa—Santok, 1 (1); Sas—Sąsiadka (1); Spl—Spławie, 2 (1); Strp—Starorypin, 1B (1); Syp—Sypniewo (1); Tum—Tum k. Łęczycy, 1 (9); Wilk—Wilkowo, 1 (1); Wol—Wolin-miasto, 1/4 (1),Wolin-miasto, 1/5 (8), Wolin—Srebrne Wzgórze, 5/II (1), Wolin, 3 (8); Wro—Wrocław–Ostrów Tumski 1 (8); Zer—Żerniki, 34 (6); Zol Żółte, 33 (4) (prepared by D. Makowiecki, drawing M. Skrzatek).

**Figure 3 animals-12-02282-f003:**
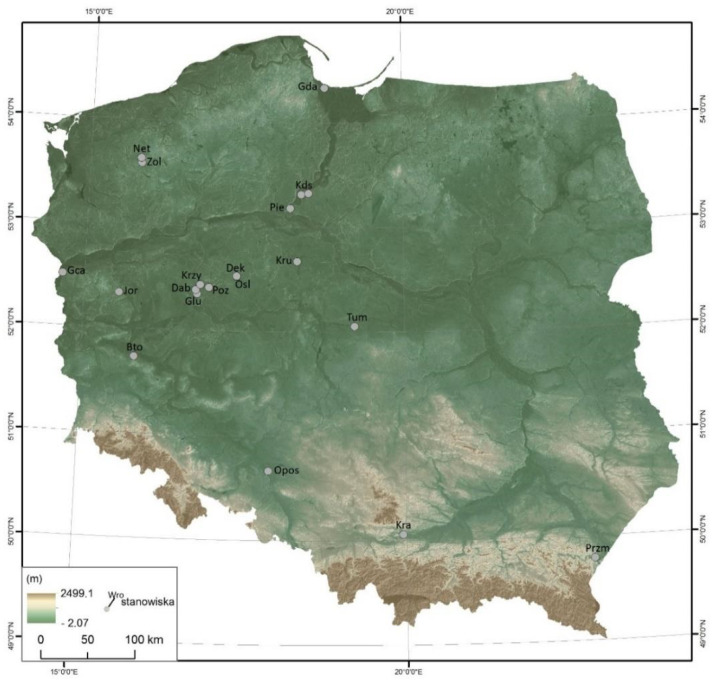
Sites with finds of horse skeletons (the acronym of the position is given in the first place, the number or name of the site in the following place, the category of the site and the number of skulls in brackets): Bto—Bytom Odrzański, 1 (stronghold—2); Dab—Dąbrówka, 2 (stronghold—2); Dek—Dziekanowice, 22 (settlement/cemetery—1); Gda—Gdańsk, Grodzka street (stronghold—8); Głu—Głuchowo, 1 (settlement—11); Gca—Górzyca, 20 (cementary 1); Jor—Jordanowo, 7 (cemetery—1); Kds—Kałdus, 3 (stronghold—9/21), Kałdus, 4 (settlement/cemetery—2); Kra—Kraków-Sukiennice (cemetery/settlement – 1); Kru—Kruszwica, 2 (stronghold, part S—1); Krzy—Krzyżowniki, 16 (settlement—1); Net—Nętno, 1 (stronghold—1); Opos—Opole-Ostrówek (stronghold—1), Osl—Ostrów Lednicki-Rybitwy, 3a (Poznań bridge—3), Ostrów Lednicki, 1 (stronghold—1), Ostrów Lednicki, 2 (borough—2); Pie—Pień, 1 (cemetery—1); Poz—Poznań-Zagórze (Alumnat) (stronghold—2); Przm—Przemyśl, 83 (cemetery—2); Sad—Sandomierz, 1 (settlement–town—1); Tum—Tum near Łęczyca, 1 (stronghold—4); Zol—Żółte, 33 (crannog-island—5) (prepared by D. Makowiecki, drawing by M. Skrzatek).

**Figure 4 animals-12-02282-f004:**
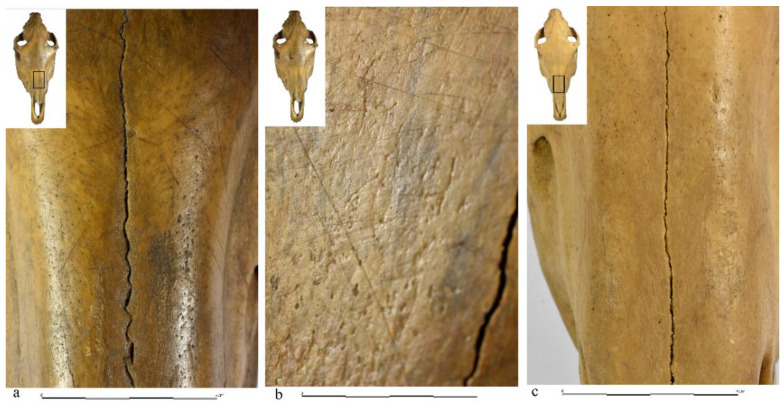
Two horse skulls of males from the Gdańsk stronghold (Grodzka street, site 1). (**a**,**b**) scratches on the bone surface; (**c**) no scratches on the bone surface. (**a**,**b**) scratches on the bone surface; (**c**) no scratches on the bone surface (photo by M. Gembicki).

**Figure 5 animals-12-02282-f005:**
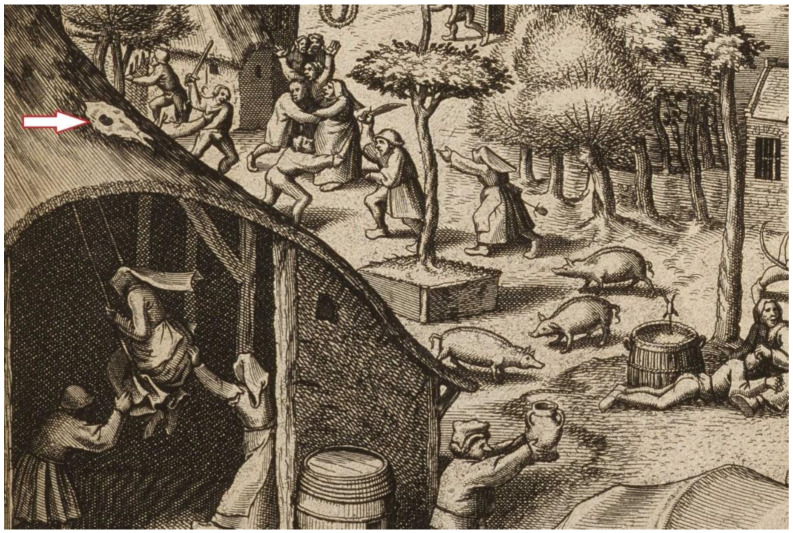
Skull of a horse on the roof of the building. St. George’s Day, painting by P. Bruegel (the part of the painting by https://upload.wikimedia.org/wikipedia/commons/3/31/Pieter_Bruegel_the_Elder_-_The_Fair_of_Saint_George%27s_Day_-_Google_Art_Project.jpg, accessed date on 5 April 2022).

**Figure 6 animals-12-02282-f006:**
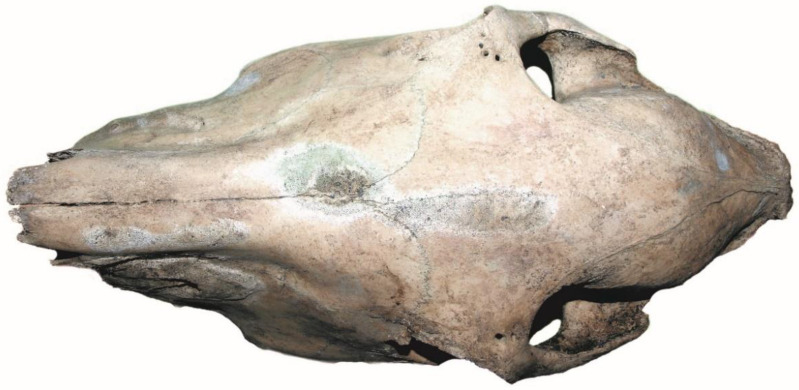
Gdańsk, Sukiennicza street (site 1). A horse’s skull with traces of a past bacterial disease (photo by D. Makowiecki).

**Figure 7 animals-12-02282-f007:**
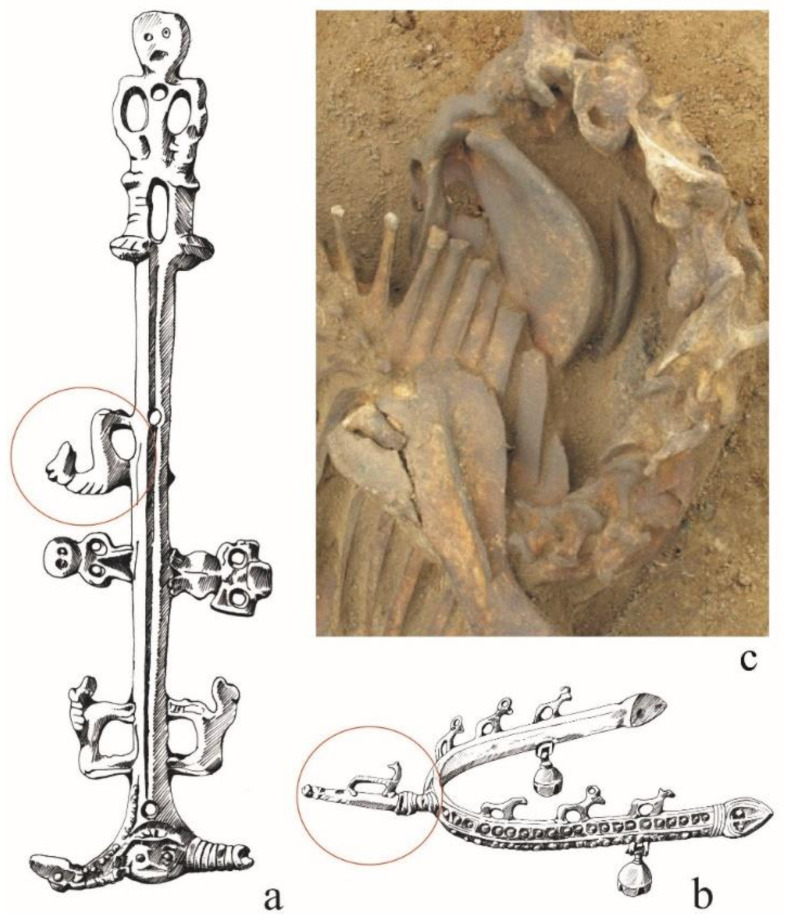
Cases of an unnatural position of the horse’s neck—discussed in the text: (**a**) the Oldenburg knife scabbard ferrule according to Kaczmarek [116]; (**b**) Spur from Ciepłe, according to Gardeła et al. [111]; (**c**) horse burial from Pień, according to Makowiecki and Janeczek [96] (Figure prepared by J. Sawicka and P. Szczepanik).

**Figure 8 animals-12-02282-f008:**
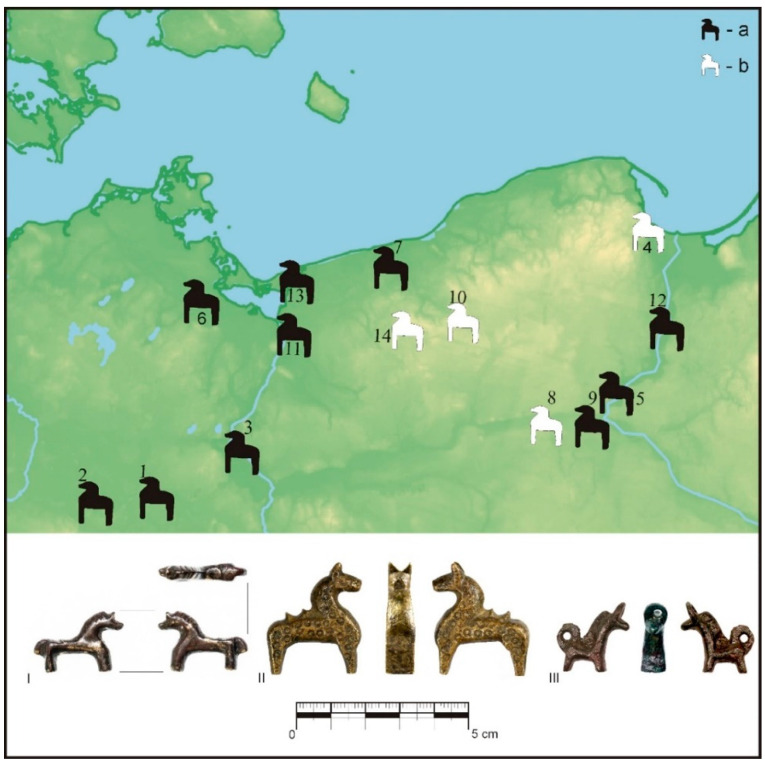
Sites with miniature horse figurines discovery in the northwest Slavic region, types of material according to Szczepanik [103]: (a) black—copper alloys; (b) white—wood; (1) Berlin-Spandau, (2) Brandenburg; (3) Cedynia, (4) Gdańsk, (5) Kałdus, (6) Krien, (7) Kołobrzeg-Budzistowo, (8) Nakło at Noteć, (9) Pawłówek, (10) Parsęcko, (11) Szczecin-Podzamcze, (12) Tymawa, (13) Wolin, (14) Żółte; (drawing by P. Szczepanik).

## Data Availability

Not applicable.

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
