# Peer review of "Horses in the Early Medieval (10th–13th c.) Religious Rituals of Slavs in Polish Areas—An Archaeozoological, Archaeological and Historical Overview"

_animals, 2022, doi:10.3390/ani12172282_

Round 1

Reviewer 1 Report

This is well prepared paper, that presents interesting data resulting from an archaeozoological, archaeological and historical overview of Early Medieval horses of Slavs in Polish areas. A few comments and corrections that I would like to suggest to the authors are as follows:

1. Figure 1 is unclear and misleading: the Prussians are Balts, not Slavs, and Rus' is not West Slavs but East Slavs.  The figure should be corrected accordingly or this explained in the captions.

2. Line 411: shouldn't it be Fig. 3 instead of Fig. 2?

3. Comment to a paragraph lines 410-436. The custom of burying horses in separate pit within a cemetery with human graves  was also prevalent in the Baltic tribes in the 3rd-14th c. AD. 

Author Response

Thank you very much for your comments. We corrected the article according to them. In the attached file are our explanations.

Reviewer 2 Report

This is an interesting study on the role of horses in early medieval Polish religious rituals. The authors combine information from written and archaeological sources.  The latter is the most relevant aspect of this work since zooarchaeological studies focused on these horses are scarce. The authors make a good contextualization of the problem and have made a great documentation work at documentary level and also for the location of the archaeological remains and their subsequent study. However, there are some aspects related to the data of the zooarchaeological and taphonomic study of the remains that need to be improved and completed in order to recommend their publication. These data are necessary to truly highlight the zooarchaeological and taphonomic study of the project, as the authors intend to do with this article.

Section 2. Materials and methods: This section has hardly any information. Part of it appears in the results and discussion section, for example, in point 3.3. We recommend expanding the section on materials and methods with a brief description of the remains studied and the sites of origin. Different methodological aspects should also appear: references used for age determination, criteria used for taphonomic identification and the study of paleopathologies. In sections 3.4 and 3.5, the age of at least two inviduals is mentioned, but the reader does not know the criteria used to establish it.

Section 3. Results and discussion:

Lines 275-277 the authors comment, "Taking into account the state of their preservation, three basic categories have been distinguished: a) loose skeletal elements (not articulated), b) complete skeletons or anatomical assemblies (with or without an anatomical system), c) skulls with mandibles, only the skulls and only the mandibles." When they talk about category "a" (loose skeletal elements) are they assuming that the horses were buried completely and that the elements have been lost? Why do they assume that this is a preservation problem? What kind of preservation problem: e.g. taphonomic, excavation or conservation malpractice on the part of museums? Could it not be that only specific parts of the animal were buried, or do they mean that the animal was buried whole but has appeared disarticulated?

On the other hand, the authors do not mention which anatomical elements of the postcranial skeleton are found when partial skeletons are found?

Lines 234-237: the authors mention that they have found horse remains from 318 sites, they have only found complete or almost complete skulls in 49 sites. It would be very interesting if the authors could provide a list of the 318 sites located. On the other hand, I would recommend that this information appear in the "materials and methods" section.

Section 3.4 "Taphonomic analyses":

The authors give some examples of taphonomic alterations identified but it would be interesting if they provided a general list of alterations, the anatomical elements where they have found them and the interpretation of the process that caused them. These could be presented in a table or in a figure using vectorized skeletons, photos or 3D images (see as an example Taylor et al., 2020).

As mentioned above, a methodological paragraph explaining the alterations analyzed and the criteria used to identify them would also be useful.

Figure 4: Improve the quality of the images, they are very bright and it is difficult to identify the "scratches" referred to by the authors. On the other hand, it would be advisable to put a detailed microscopic photo of some of these "scratches".

They only give examples of the treatment of the skulls but not of the rest of the postcranial skeleton to see if it received any type of treatment. Have these studies not been carried out? There are no markings on the postcranial skeleton?

Although the examples offered by the authors are very interesting, it would also be important for them to offer a general assessment of the taphonomic analysis of the remains that would be illustrated by the table or figures mentioned above: frequency of marks, types of marks identified, and general assessment and interpretation of the marks.

Section 3.5 Paleopathological analyses

As in the previous section, it would be interesting for the authors to offer a list of pathologies, their location and the interpretation of the disease that causes them, as well as a general evaluation of the identified pathologies and their interpretation.

Lines 433-435: "In all cases, it can be considered that the location of the burials of such horses in the cemetery would be characterized by intentions of a non-utilitarian, and therefore magical, symbolic, and at least emotional basis". Animals can participate in the funerary ritual in many different ways: as psychopomps, offerings, culinary offerings, part of feasts...(e.g., Collins, 1990; De Grossi Mazzorin 2008 ; Serjeantson 2011). In this case, we assume that the authors refer to complete skeletons, in anatomical connection and without evidence of processing or consumption. It would be important for the authors to specify this because otherwise the interpretations of the presence of these animals in these graves may be multiple.

References:

Collins BJ (1990) The puppy in Hittite ritual. J Cuneif Stud 42(2):211–226.

De Grossi Mazzorin J (2008) L'uso dei cani nel mondo antico nei riti di fondazione purificacione e passaggio. In: d’Adria F, de Grossi J, Fiorentino G (eds) Uomini, piante e animali nella dimensione del sacro. Edipuglia, Lecce, pp 71–82 226.

Serjeantson D (2011) Review of animal remains from the Neolithic and early bronze age of southern Britain, Research Department Report Series 29. English Heritage, Portsmouth

Taylor, W., Fantoni, M., Marchina, Ch., Lepetz, S., Bayarsaikhan, J., Houle, J.-L., Pham, V., Fitzhugh, W., (2020). Horse sacrifice and butchery in Bronze Age Mongolia. J Archaeol Sci: Reports.  https://doi.org/10.1016/j.jasrep.2020.102313

Author Response

Thank you very much for all your comments given to us. We appreciate them all very much. Most of them we used for improving our article, but not all. Please, note that the article has some problems, and there is no chance to present all relevant data, which are still collected, with their analysis, discussion and conclusions in one article. Below are placed explanations of your comments with indications of what was corrected and what is, in the current stage of our research, impossible but will be the subject of following articles.

Please, see the attachment. In it, we indicate all changes and our explanation concerning your comments. 
